# Putting the Personalized Metabolic Avatar into Production: A Comparison between Deep-Learning and Statistical Models for Weight Prediction

**DOI:** 10.3390/nu15051199

**Published:** 2023-02-27

**Authors:** Alessio Abeltino, Giada Bianchetti, Cassandra Serantoni, Alessia Riente, Marco De Spirito, Giuseppe Maulucci

**Affiliations:** 1Neuroscience Department Biophysics Section, Università Cattolica del Sacro Cuore, 00168 Rome, Italy; 2Fondazione Policlinico Universitario A. Gemelli IRCSS, 00168 Rome, Italy

**Keywords:** metabolism, deep learning, gated recurrent unit, long short-term memory, transformer, wearables, forecasting, diet plans, digital nutrition, digital twin, SARIMAX

## Abstract

Nutrition is a cross-cutting sector in medicine, with a huge impact on health, from cardiovascular disease to cancer. Employment of digital medicine in nutrition relies on digital twins: digital replicas of human physiology representing an emergent solution for prevention and treatment of many diseases. In this context, we have already developed a data-driven model of metabolism, called a “Personalized Metabolic Avatar” (PMA), using gated recurrent unit (GRU) neural networks for weight forecasting. However, putting a digital twin into production to make it available for users is a difficult task that as important as model building. Among the principal issues, changes to data sources, models and hyperparameters introduce room for error and overfitting and can lead to abrupt variations in computational time. In this study, we selected the best strategy for deployment in terms of predictive performance and computational time. Several models, such as the Transformer model, recursive neural networks (GRUs and long short-term memory networks) and the statistical SARIMAX model were tested on ten users. PMAs based on GRUs and LSTM showed optimal and stable predictive performances, with the lowest root mean squared errors (0.38 ± 0.16–0.39 ± 0.18) and acceptable computational times of the retraining phase (12.7 ± 1.42 s–13.5 ± 3.60 s) for a production environment. While the Transformer model did not bring a substantial improvement over RNNs in term of predictive performance, it increased the computational time for both forecasting and retraining by 40%. The SARIMAX model showed the worst performance in term of predictive performance, though it had the best computational time. For all the models considered, the extent of the data source was a negligible factor, and a threshold was established for the number of time points needed for a successful prediction.

## 1. Introduction

Over the past few decades, precise diagnosis and personalized treatment have become increasingly important in healthcare [1]. Nutrition, as an important factor of personalized treatments, has a huge impact on health, from cardiovascular disease to cancer [2,3]. Nutritional habits have been linked to stronger immunity, a lower risk of noncommunicable diseases (such as diabetes and cardiovascular disease) and increased life expectancy [4,5].

Increased knowledge of the effects of nutrition on pathophysiologies of diseases, achieved with new diagnostic and monitoring technologies spanning from -omics [6,7] to wearable devices [8], has offered innovative solutions for personalized treatments. Among the most striking innovations, digital twins (DTs), which are digital replicas of human physiology, represent an emergent solution for prevention and treatment of many diseases [9,10]. DT technology holds the promise of starkly reducing the cost, time and manpower required to test effects of dietary and physical-activity plans, to run clinical trials and to create personalized diets for citizens and patients. DT models are built on data flows sourced from connected biomedical devices on the Internet of Things (IoT) and collected through digital web-based applications integrating dietary, anthropometric and physical activity data, such as the one developed by our research group [11]. Artificial intelligence algorithms have shown good performance in analysis of biometric signals [12,13]. The data streams provided by these data acquisition platforms can be analyzed with data-driven models of human metabolism, such as the personalized metabolic avatar (PMA) [14] developed by our group, to estimate personalized reactions to diets. The PMA consists of a gated recurrent unit (GRU) deep-learning model trained to forecast personalized weight variations according to macronutrient composition and daily energy balance. This model can perform simulations and evaluations of diet plans, allowing definition of tailored goals for achieving ideal weight. However, putting PMAs into production and transforming them in a reliable, fast and continuously updating model for predictive analytics is a difficult task. Among the principal issues, challenges can arise from changes to data sources, models and model parameters, which introduce room for error and overfitting and can lead to abrupt variations in computational time. To overcome these issues, here, we selected the best strategy for deployment in terms of predictive performance and computational time. Among statistical models, we selected, as a representative, the SARIMAX (Seasonal Auto-Regressive Integrated Moving Average with eXogenous factors) model: the most complete for multivariate forecasting. Among deep-learning models, we selected recurrent neural networks (RNNs), such as gated recurrent units (GRUs) [14] and long short-term memory (LSTM) networks, and a new model recently introduced, the Transformer model [15], which has shown great results both in natural language processing and in time-series forecasting [16]. Moreover, we have tested the influence of the data number retrieved, which, in real settings, can vary in range from user to user, on the models. These efforts are necessary to put these models into production to augment citizens’ self-awareness, with the aim of achieving long-lasting results in pursuing a healthy lifestyle.

## 2. Materials and Methods

### 2.1. Study Population

In this single-arm, uncontrolled-pilot prospective study, a group of 10 voluntary adults (60% females and 40% males, 3 overweight and 7 normal), recruited among our lab staff, self-monitored daily their weight, diet and activities completed for at least 100 days, as explained in a previous work [11]. The participants shared their personal data after signing informed consent. 

### 2.2. Wearables and Devices 

To track anthropometric data, the following devices were used: The MiBand 6, a smartband of Xiaomi Inc.^®^ (Beijing, China), for estimating calories burned during exercise (walking, running, etc.).The Mi Body Composition Scale, an impedance balance of Xiaomi Inc.^®^ (Beijing, China), for tracking weight and RMR.

These devices were already used in 4 studies on PubMed, and 11 clinical trials have been performed using the MiBand1. Validation results in estimating RMR can be retrieved in recent publications [4]. For tracking the food diary for each participant, a website app (ArMOnIA, https://www.apparmonia.com, accessed on 7 February 2023) developed by our group was used for the storing of food data. These data had already been validated in two other studies [11,14].

### 2.3. Datasets

As already shown in [14], for the development of the deep-learning models implementing PMAs, the following data were used:
$var_{1}$: Weight: w(t) [kg] $var_{2}$: Energy Balance (EB): Eb(t) [kcal] $var_{3}$: Carbohydrates: $m_{c}\left(t\right)$ [g]$var_{4}$: Proteins: $m_{p}\left(t\right)$ [g]$var_{5}$: Lipids: $m_{l}\left(t\right)$ [g]

Where $var_{j}$ stands for variable *j*, with *j* = 1, …, 5.

In Figure 1a,b, the representative time series of the five selected quantities are reported.

We reframed the time-series forecasting problem as a supervised learning problem, using lagged observations (including the seven days before the prediction, e.g., *t* − 1, *t* − 2, *t* − 7) as input variables to forecast the current time step (*t*), as already explained in [12]. The inputs of our model were $var_{1}\left(t-7\right),\;\ldots ,\;var_{5}\left(t-k\right),\;\ldots ,\;var_{j}\left(t-i\right),\;\ldots ,\;var_{5}\left(t-1\right)$, with *i* = 1, …, 7 indicating the lagged observation and *j* = 1, …, 5 indicating the input variable. Therefore, the total number of inputs for the PMA was 7∙5 = 35. In this notation, the output of the PMA is $var_{1}\left(t\right)$, i.e., the weight at time *t*. 

The dataset fed to the SARIMAX model is described in the next section.

### 2.4. Description of Models

As explained in the introduction, DDMs are divided into two types: statistical and deep-learning models. To select the best option for the development of the PMA, we chose to compare 4 different models:

SARIMAX:

The SARIMAX model (Seasonal Auto-Regressive Integrated Moving Average with eXogenous factors) is a linear regression model: an updated version of the ARIMA model. It is a seasonal equivalent model, like the SARIMA (Seasonal Auto-Regressive Integrated Moving Average) model, but it can also deal with exogenous factors, which are accounted for with an additional term, helping to reduce error values and improve overall model accuracy. This model is usually applied in time-series forecasting [17].

The general form of a *SARIMA*(*p*,*d*,*q*)(*P*,*D*,*Q*,*s*) model is
(1)$$\Theta {\left(L\right)}^{p}\theta {\left(L^{s}\right)}^{p}\Delta^{d}\Delta_{s}^{D}w_{t}=\Phi {\left(L\right)}^{q}\varphi {\left(L^{s}\right)}^{Q}\Delta_{s}^{D}\epsilon_{t}$$
where each term is defined as follows:
$\Theta {\left(L\right)}^{p}$ is the nonseasonal autoregressive lag polynomial;$\theta {\left(L^{s}\right)}^{p}$ is the seasonal autoregressive lag polynomial;$\Delta^{d}\Delta_{s}^{D}w_{t}$ is the time series, differenced d times and seasonally differenced D times;$\Phi {\left(L\right)}^{q}$ is the nonseasonal moving average lag polynomial;$\varphi {\left(L^{s}\right)}^{Q}$ is the seasonal moving average lag polynomial.

When dealing with n exogenous values, each defined at each time step, *t*, denoted as $x_{t}^{i}$ for i≤n, the general form of the model becomes
(2)$$\Theta {\left(L\right)}^{p}\theta {\left(L^{s}\right)}^{p}\Delta^{d}\Delta_{s}^{D}w_{t}=\Phi {\left(L\right)}^{q}\varphi {\left(L^{s}\right)}^{Q}\Delta_{s}^{D}\epsilon_{t}+\sum_{i=1}^{n}\beta_{i}x_{t}^{i},$$
where $\beta_{i}$ is an additional parameter accounting for the relative weight of each exogenous variable.

In Appendix A, additional details about the model are reported.

We implemented this model on Python using the StatsModels library (https://www.statsmodels.org/stable/index.html, accessed on 7 February 2023), with the SARIMAX (https://www.statsmodels.org/0.9.0/generated/statsmodels.tsa.statespace.sarimax.SARIMAX.html, accessed on 7 February 2023) class.

For the SARIMAX model, $var_{2}$, $var_{3}$, $var_{4}$ and $var_{5}$ (i.e., *EB* and macronutrients) are considered as exogenous variables, with the weight as output. Considering our dataset structure, the exogenous variables at time *t* correspond to the inputs for the forecasting of the weight at time *t* + 1. However, in the SARIMAX equation, the exogenous term is considered at the same time, t, with respect to the output. To overcome this issue, we shifted the exogenous values of ΔT = 1 day with respect to weight. In this way, the exogenous term changed as follows: $\sum_{i=1}^{n}\beta_{i}x_{t-1}^{i}$.

LSTM:

Long short-term memory (LSTM) networks [18], a variant of the simplest recurrent neural networks (RNNs), can learn long-term dependencies and are the most widely used for working with sequential data such as time-series data [19,20,21].

The LSTM cell (Figure 2) uses an input gate, a forget gate and an output gate (a simple multilayer perceptron). Depending on data’s priority, these gates allow or deny data flow/passage. Moreover, they enhance the ability of the neural network to understand what needs to be saved, forgotten, remembered, paid attention to and output. The cell state and hidden state are used to gather data to be processed in the next state.

The gates have the following equations:Input Gate:
(3)$$i=\sigma \left(W_{i}h_{t-1}+W_{i}h_{t}\right),$$Forget Gate:(4)$$f=\sigma \left(W_{f}h_{t-1}+W_{f}h_{t}\right),$$Output Gate:(5)$$o=\sigma \left(W_{o}h_{t-1}+W_{o}h_{t}\right),$$Intermediate Cell State:(6)$$g=tanh\left(W_{g}h_{t-1}+W_{g}h_{t}\right),$$Cell State (Next Memory Input):(7)$$c_{t}=\left(g*i\right)+\left(f*c_{t-1}\right),$$New State:(8)$$h_{t}=o*tanh\left(c_{t}\right),$$
with $X_{t}$ as the input vector, $h_{t}$ as the output vector, *W* and *U* as parameter matrices and f as the parameter vector.

We implemented the LSTM network using the TensorFlow Keras library (https://www.tensorflow.org/api_docs/python/tf/keras, accessed on 7 February 2023), which implements an LSTM cell as an available class on Python (https://www.tensorflow.org/api_docs/python/tf/keras/layers/LSTM, accessed on 7 February 2023), which we added into a model as a monolayer neural network.

GRU:

The gated recurrent unit, just like the LSTM network, is a variant of the simplest RNN but with a less complicated structure. It has an update gate, *z*, and a reset gate, *r*. These two variables are vectors that determine what information passes or does not pass to output. With the reset gate, new input is combined with the previous memory while the update gate determines how much of the last memory to keep. 

The GRU has the following equations:Update Gate:
(9)$$z=\left(W_{z}h_{t-1}+U_{z}x_{t}\right),$$Reset Gate:(10)$$r=\left(W_{r}h_{t-1}+U_{r}x_{t}\right),$$Cell State:(11)$$c=tanh\left(W_{c}\left(h_{t-1}*r\right)+U_{c}x_{t}\right),$$New State:(12)$$h_{t}=\left(z*c\right)\left(\left(1-z\right)*h_{t-1}\right),$$

A GRU cell is shown in Figure 3.

A more accurate description can be found in the Appendix A of a previous work [14].

As for the LSTM network, we implemented the GRU in TensorFlow using a GRU cell (https://www.tensorflow.org/api_docs/python/tf/keras/layers/GRU, accessed on 7 February 2023) implemented into a monolayer neural network. 

Transformer:

LSTM and GRUs have been strongly established as state-of-the-art approaches in sequence modeling and transduction problems such as language modeling and machine translation [22,23,24,25,26] because of their ability to memorize long-term dependency. Since they are inherently sequential, there is no parallelization within training examples, which makes batching across training examples more difficult as sequence lengths increase. Therefore, to allow modeling of dependencies for any distance in the input or output sequences, attention mechanisms have been integrated in compelling sequence modeling and transduction models in various tasks [24,27]. Commonly [28], such attention mechanisms are used in conjunction with a recurrent network. In 2017, a team at Google Brain^®^ developed a new model [15], called “Transformer”, with an architecture that avoids recurrence and instead relies entirely on an attention mechanism to draw global dependencies between inputs and outputs. This architecture uses stacked self-attention and pointwise, fully connected layers for both the encoder and the decoder, shown in the left and right halves of Figure 4, respectively. In Appendix A, a more accurate description of the model is reported.

The implementation of the model in Python followed the Transformer starting code shared by the Google Brain team (https://keras.io/examples/timeseries/timeseries_transformer_classification/, accessed on 7 February 2023). 

### 2.5. Model Selection and Comparison

#### 2.5.1. Implementation and Selection of Models

For each selected model, parameter scanning was performed, and the best model was selected. Below, procedures are indicated according to models.

SARIMAX:

Augmented Dickey–Fuller (ADF) tests, applied to a weight time series, yielded *p*-values larger than α = 0.05 for 90% of the overall participants. Therefore, we transformed the weight time series into a stationary one that performed first-order differentiation. The ADF test, repeated on preprocessed series, confirmed stationarities for all of the transformed time series. Following this adjustment, the terms *d* and *D* were each set to 1. 

We started with fitting a SARIMAX model for all the datasets available, considering the ranges in Table 1.

In the literature, the most common way to find the best parameters for SARIMAX models is based on a simple grid search following the Akaike information criterion (AIC) and the Bayesian information criterion (BIC), respectively. These criteria help to select the model that explains the greatest amount of variation using the fewest possible independent variables, using maximum likelihood estimation (MLE) [29], and they both penalize a model for having increasing numbers of variables, to prevent overfitting.

Therefore, we ranked the models according to the lowest AIC values. The first 5 models were then trained on the datasets, and the root mean squared error (RMSE) scores were calculated. The model with the lowest RMSE was then selected.

The LSTM and GRU Models:

Hyperparameter tuning with the aim of minimizing loss function was carried out to select the best deep-learning model [14]. Typically, in time-series forecasting, tuning is carried out to reduce the RMSE of test-training forecasting. 

Considering that the LSTM and GRU models had the same configuration and the same hyperparameters, we proceeded with both to parameter scanning in the range shown in Table 2.

We selected the best model via considering the lowest RMSE obtained from a prediction on the same training-test sets.

Transformer:

The implementation of this model into Keras was like that of the other two neural networks, with some exceptions for the hyperparameters. We considered a grid search that would take into account the range of the hyperparameters shown in Table 3. 

Differently from LSTM and GRU, there are two more parameters: head size, which is the dimensionality of the query, key and value tensors after the linear transformation, and num heads, which is the number of attention heads.

In this case, we also chose the best model via minimizing the RMSEs on the training-test sets.

#### 2.5.2. Performance of Models with Datasets of Varying Length

Following model selection and parameter optimization, we compared the models, considering, as a quality index, the RMSE, which indicates errors in weight prediction with a test-set length of 7 days, considering a training set of more than 100 days (mean ± SD = 161.3 ± 22.4) for each participant.

In addition, since scarcity of data is a common problem in deployment of PMAs in production, we tested the models in more realistic settings. We thus divided the dataset of each participant into 9 independent groups of 15 days. Then, we evaluated the RMSE on a test set with a length of 1 day for each group (with a training set of 14 elements). The final RMSE was the average of these 9 RMSEs. An ANOVA followed by a Tukey test was applied for pairwise comparison of RMSEs.

#### 2.5.3. Computational Time

In addition to prediction performance, the computational times were calculated for the retraining and prediction phases for the four models. 

A Kruskal–Wallis test followed by a Dunn test was applied for pairwise comparison of computational times.

### 2.6. Computational Requirements and Python Libraries

Computational requirements were minimal in order to allow deployment on virtual machines available on the web. The code for the development of the models was run in Google Colab with the default settings (free plan). The code requires the following libraries: *tensorflow* = 2.9.2 (https://pypi.org/project/tensorflow/, accessed 7 February 2023), *pandas* = 1.3.5 (https://pandas.pydata.org/, accessed 7 February 2023), *numpy* = 1.21.6 (https://numpy.org/, accessed 7 February 2023), *matplotlib* = 3.2.2 (https://matplotlib.org/, 7 February 2023), *seaborn* = 0.11.2 (https://seaborn.pydata.org/, accessed 7 February 2023), *statsmodels* = 0.12.2 (https://www.statsmodels.org/stable/index.html, accessed on 7 February 2023), *scipy* = 1.7.3 (https://pypi.org/project/scipy/, accessed on 7 February 2023), *bioinfokit* = 2.1.0 (https://pypi.org/project/bioinfokit/0.3/, accessed on 7 February 2023), *scikit-learn* = 1.0.2 (https://scikit-learn.org/stable/, accessed 7 February 2023) and *scikit-posthocs* = 0.7.0 (https://scikit-posthocs.readthedocs.io/en/latest/, accessed on 7 February 2023).

## 3. Results

### 3.1. Selection of the Optimal Model

We started with optimizing parameters for each selected model and each participant, as explained in par. 2.6.1.

For the GRU, LSTM and Transformer models, we considered an Adam optimizer and, as a loss function, the mean absolute error (MAE), defined with the formula
(13)$$MAE=\frac{\sum_{i=1}^{n}\left|y_{i}-x_{i}\right|}{n},$$
where $y_{i}$ is the actual value and $x_{i}$ is the prediction.

In Appendix A, the selected parameters for each user are reported for each type of model. As shown in [14], we trained a model for each user to adapt it to the personalized characteristics of metabolism.

### 3.2. Comparison between Models

As explained in par. 2.6.2, to compare model performance, we used the RMSE of the prediction of the test set for each participant. Datasets were structured to make the training and test set homogeneous, ensuring that the models learned from the same data and tested their knowledge under identical conditions. In Figure 5, we report the forecasting with each model for a single participant.

From a visual inspection, the GRU and LSTM models follow the variations of weight more accurately while the SARIMAX model shows the worst result. In Figure 6, we show the RMSEs, grouped based on models, for each participant. Indeed, there is an evident difference between the SARIMAX model and the others, confirming that neural networks outperform statistical models in time-series forecasting. On the other hand, the deep-learning models show comparable RMSEs to each other.

Hence, the Transformer model did not demonstrate improvement with respect to the GRU or LSTM models, having, on the contrary, slightly worse results.

To quantify these observations, we carried out an ANOVA among the RMSEs of the models, showing a *p*-value lower than $\alpha =0.05\;\left(4.31\cdot 10^{-4}\right)$ and confirming that there was at least one model different from the others. We then performed a Tukey test for pairwise comparison, and the results, reported in Table 4, confirmed that the SARIMAX model is different from the others (adjusted *p*-value lower than α), while there is no statistical difference among the other three models, yielding a *p*-value bigger than α. 

### 3.3. Analysis of the Performance with a Limited Dataset Length

As explained in Section 2.6, PMAs often operate on datasets with limited length. For example, diet diaries are often compiled for a limited amount of time. Therefore, we carried out a test to show the performances of these models, considering a limited dataset of 15 days. The model was trained to predict the weight for the day afterward.

To acquire a reliable index of the performance of each model, we tested the models on nine subsets of data in the original dataset for each participant. In this way, we could refer to a mean RMSE for each model and for each participant.

In Figure 7, the RMSE distribution of each model is reported (each point represents a user). From a visual inspection, we can conclude that, again, the SARIMAX model displays the worst results, while the others have similar performances. To confirm this observation, we carried out an ANOVA (*p*-value = 0.019) followed by a Tukey test. The pairwise comparison showed that only the SARIMAX model presented accuracy that was statistically different from that of the deep-learning models.

### 3.4. Performance versus Data Length

The results reported in the previous section show how the model provided accurate solutions for few data. In this section, we analyze changes in performance with decreasing data length. We considered the following subsets: 100% of the dataset and 100, 80, 60, 40 and 30 days. In Figure 8, we report the RMSE versus the data length for each participant and for each model, with error bars representing the standard deviations (SDs).

While the SARIMAX model showed an important decrease in performance as data length decreased, the others were characterized with stable performances, also with data collected only for thirty days.

### 3.5. Computational Time

In the evaluation of the performance of a model in a production environment, we must consider another important parameter: the computational cost, expressed in time. This computational time is the sum of the (re)training time and the forecasting time, since in a production environment, the model must be retrained every time and data are gathered in real time. In Table 5, we report, for each model, the computational time, the (re)training time and the forecasting time. The times are averaged based on the number of participants.

It is possible to observe how the GRU and LSTM models each require about 1/5 of the time requested to retrain and forecast with the Transformer model but 10× more time than that of the SARIMAX model. Therefore, a major burden of the Transformer model resides in the retraining time, since it requires more complex operations than the others.

To quantify these observations, a Kruskal–Wallis test was carried out [30] among the models, since a Shapiro–Wilk test [30] had confirmed that distributions would not be normal. The test yielded a *p*-value < 0.05, showing the presence of a statistical difference between the models. Therefore, a posthoc test (Dunn test [31]) was carried out to investigate the pairwise comparison. The test showed no statistical difference between the GRU and LSTM models, confirming that they have similar performances (Figure 9), which are better than that of the Transformer model.

## 4. Discussion

Obesity and cardiovascular disease, as the most serious public health challenges of the 21st century, are strongly conditioned through dietary habits. Digital health can help people to monitor themselves and prevent these diseases. The advent of wearable devices and the evolution of smartphone technology have allowed the development of an infrastructure able to retrieve data that could be used for the development of what is defined as a “Digital Twin”: a digital representation of human physiology. With this technology, it is possible to import digital health into the lifestyles of citizens, promoting a healthy lifestyle, since people would be in conditions to better know their own physiologies and responses to nutrition and physical activity. Here, we relied on the ArMOnIApp application, which is able to fetch, preprocess and analyze spontaneous and voluntary physical activities (PAs), dietary measures and anthropometric measures from a set of commercial wearables and other smart devices provided to the end user [11]. These data led to the development of a model, the PMA, that is able to give personalized responses for each end user, such as personalized reactions to the introduction of a particular food in their diet [14]. Here, we compared predictive and computational performances of several models, with the aim of providing useful parameters to put the PMA into production. Moreover, we tested the influence of the data number retrieved, which, in real settings, can vary in range from user to user, on the four models. In a production environment, the practice of automating deployment, integration and monitoring of machine-learning (ML) models is called MLOps [32], and this automation is crucial to increase the speed at which organizations can release models into production. MLOps also involves ensuring continuous quality and dynamic adaptability of projects throughout the entire model lifecycle [32].

To make an efficient and accurate PMA, data must be retrieved in real time. Therefore, web applications must be structured to continuously fetch new data as they are made available with devices, and to control data quality using algorithms. To include these functionalities, we relied on our web application, ArMOnIApp [11]. Moreover, ML models require automation of model retraining, and in this framework, the time cost for this procedure has an important role to optimize end performance. To this aim, we evaluated the time necessary for retraining of and prediction for the most used and reliable forecasting models. The results (summarized in Table 6) show how the GRU and LSTM models require about 1/5 of the computational time of the Transformer model, despite this time being more than 10 times that of the SARIMAX model.

On top of these optimizations, there is a need to monitor quality of predictions. To this aim, we outlined a workflow to evaluate the performances of different models with varying data lengths. We found out that the SARIMAX model, though being the fastest, had the worst RMSE, with a great variability among users. This RMSE, being four times higher than that of the GRU or LSTM model, penalizes the SARIMAX model in the deployment of the PMA. In terms of the RMSE, the Transformer model had a better performance than the SARIMAX model as well, but was comparable with RNNs. However, the time cost was the highest (four times higher than for the GRU/LSTM model), and this criticality has a strong impact on production development.

According to the performances and computational times, we can conclude that the PMAs built on the GRU or LSTM model show optimal predictive performances with acceptable computational time, making them the best candidates for a production environment.

Another issue is the need to compare the effectiveness of training several ML models specialized in different groups versus training one unique model for all the data. To address this issue, planning to create a unique model accounting for the metabolisms of a cohort of participants will require an increased number of participants.

Before these models are put into production, several ethical concerns must be moreover addressed. In regard to privacy concerns, collection and storage of personal health data by wearable devices can potentially compromise users’ privacy if this information is shared with third parties without their consent. In this study, we retrieved health data from the Zepp API, where users have explicitly consented to data sharing. The privacy policy can be retrieved on the Zepp website (https://www.zepp.com/privacy-policy, accessed on 24 February 2023). There are, in any case, security risks. Wearable devices are often connected to the internet, making them vulnerable to hacking and data breaches. This can result in sensitive personal data being stolen or compromised, potentially leading to identity theft or other forms of fraud. In addition, discrimination is an issue to be addressed, since use of wearable devices and data collected can potentially lead to discrimination against individuals with pre-existing health conditions or disabilities. This can result in denial of insurance coverage or job opportunities. Finally, there are social implications: use of wearable devices to track personal data can promote unhealthy obsessions with self-monitoring. These issues have been constantly monitored in pilot and clinical studies, but protocols must be developed and optimized before the use of these systems on a large scale is allowed. Some of these protocols already exist or are under research [33,34].

## 5. Conclusions

Putting the PMA into production can produce diets and activity regimens that are specifically tailored to users’ needs. Thanks to the PMA, pertinent hints can be found to provide citizens and nutritionists with scientific knowledge and reliable tools, enhancing their self-awareness and assisting them in their quests for healthy lifestyles. An important development might be inclusion of newly developed lipid metabolism indicators (such as membrane lipids and fluidity of red-blood-cell membranes) as input in the PMA to research the impacts and influences of dietary components on their results [35,36,37,38,39]. Additionally, cutting-edge and promising anthropometric markers, such as VO2max and heart rate frequency, monitored using wearable technology can enhance the accuracy of weight predictions [40]. These integrations could group and cluster various PMA responses, providing insights into these variables that could affect an individual’s metabolism. Another important advancement may come from the advent of quantum computing and the achievement of quantum supremacy [41], which will revolutionize ML models, including the PMA, via increasing their performances and reducing their computational times.

## Figures and Tables

**Figure 1 nutrients-15-01199-f001:**
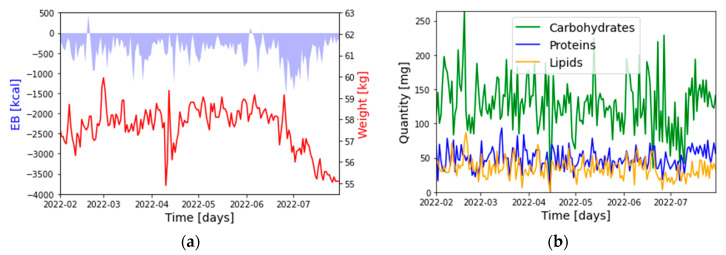
Weight, EB (**a**) and food composition (**b**) time series (for user 2).

**Figure 2 nutrients-15-01199-f002:**
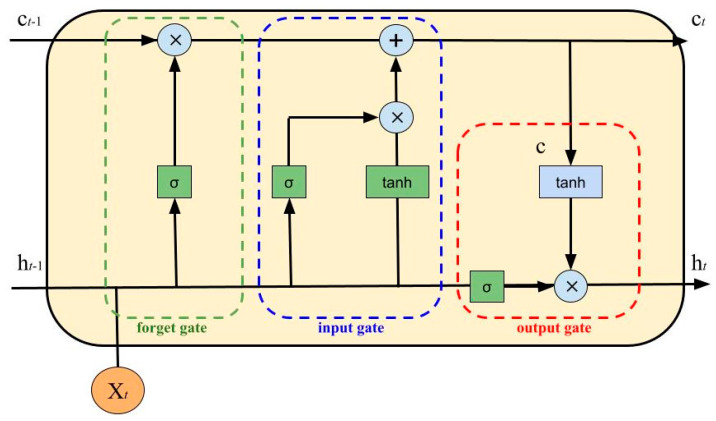
An LSTM cell, where σ is the sigmoid function.

**Figure 3 nutrients-15-01199-f003:**
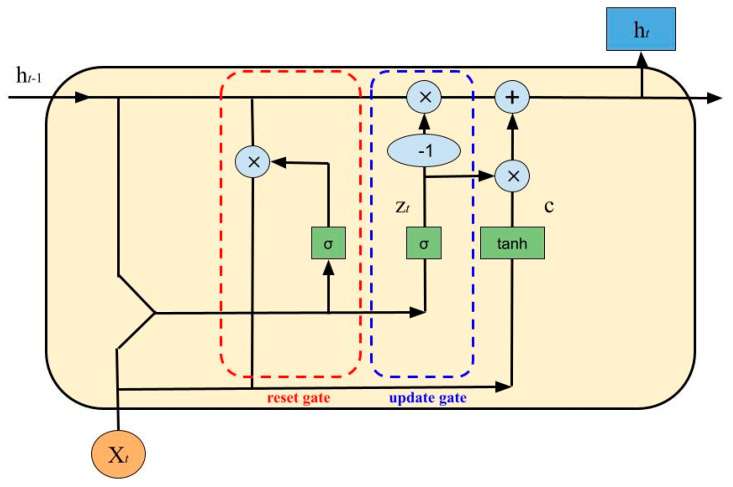
A GRU cell, where σ is the sigmoid function.

**Figure 4 nutrients-15-01199-f004:**
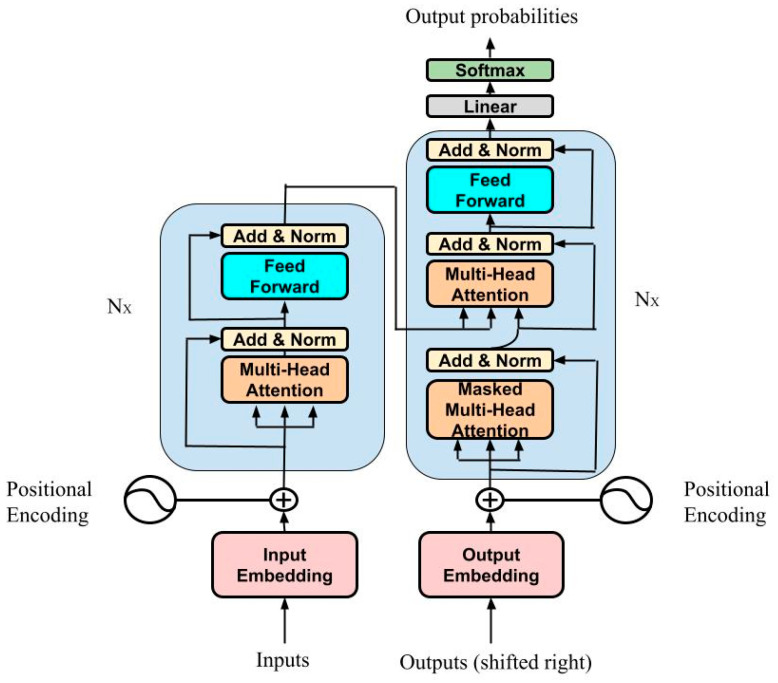
Transformer architecture. In Appendix A, an accurate description of the architecture is reported.

**Figure 5 nutrients-15-01199-f005:**
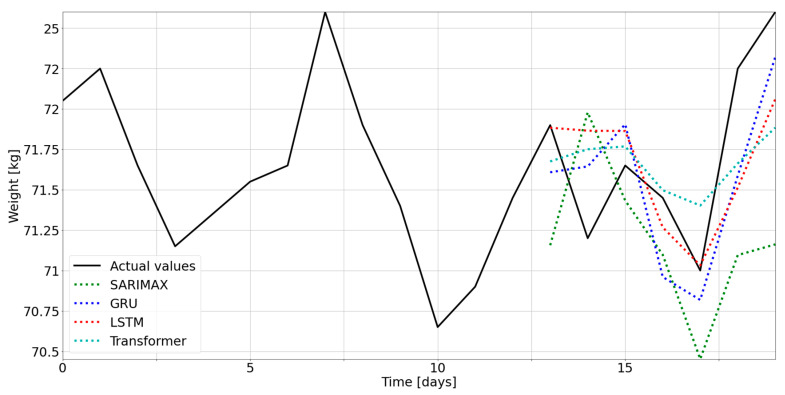
Predictions of test set of user 0, for all models.

**Figure 6 nutrients-15-01199-f006:**
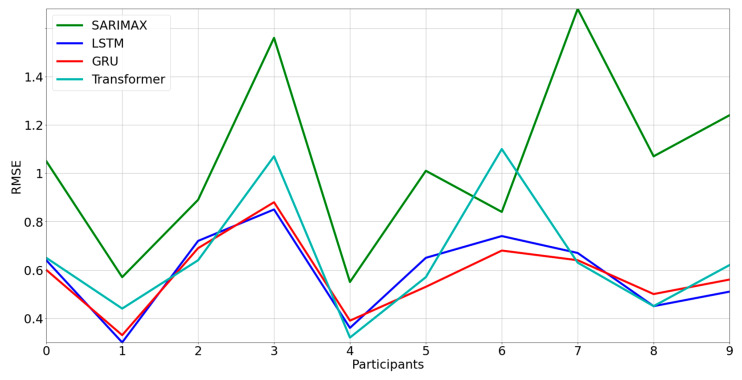
RMSEs for each model versus users.

**Figure 7 nutrients-15-01199-f007:**
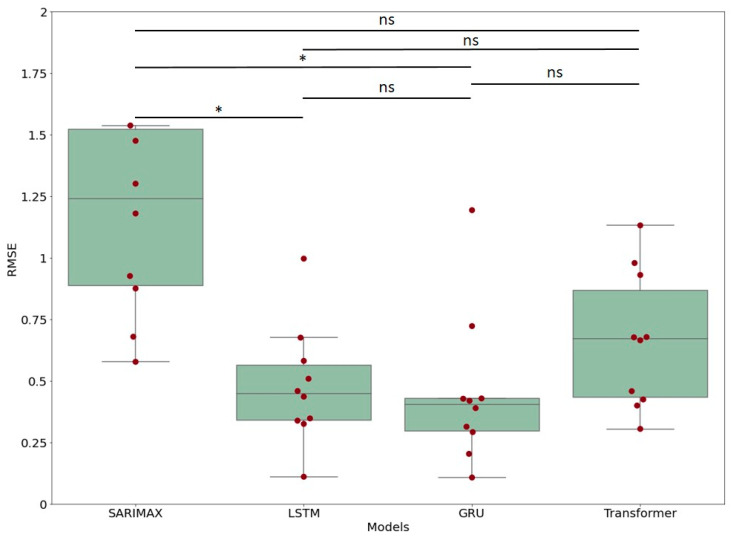
Distribution of RMSEs for each model and for each participant for a forecasting of 1 day, with a training set of 14 days for nine subsets, where * stands for a *p*-value < 0.05. ns: not significant.

**Figure 8 nutrients-15-01199-f008:**
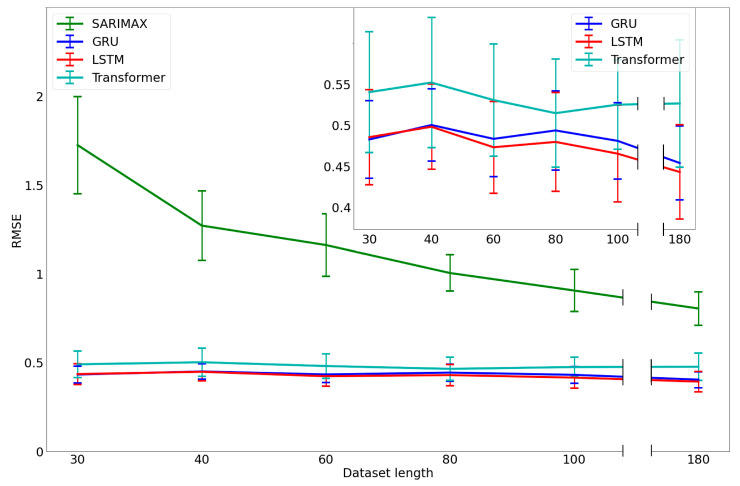
Mean RMSEs and deviation standards for each model for different dataset lengths, with a 7-day forecasting. In the inset at right, an enlargement of the deep-learning RMSEs is reported.

**Figure 9 nutrients-15-01199-f009:**
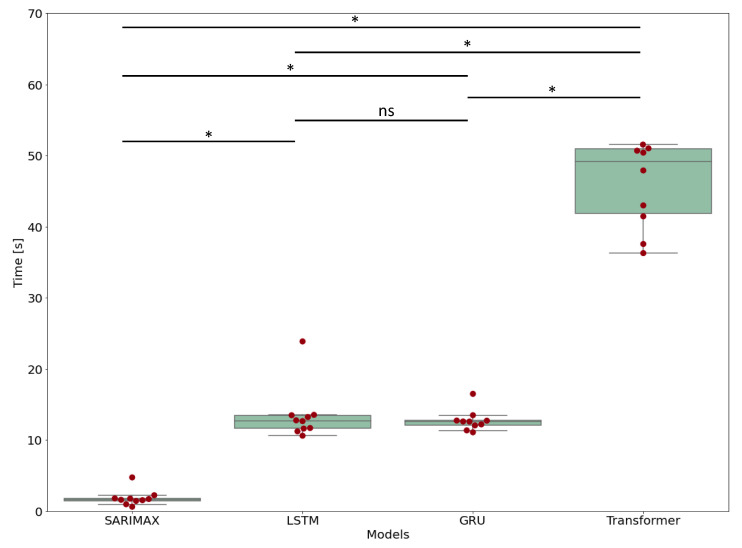
Computational time distribution for each model, with pairwise comparison, where * stands for a *p*-value < 0.05. ns: not significant.

**Table 1 nutrients-15-01199-t001:** Parameter range for the selection of the best SARIMAX model.

p	q	d	P	Q	D	S
(1–9)	(1,2)	(1,2)	(1–5)	(1,2)	(1,2)	(7)

**Table 2 nutrients-15-01199-t002:** Hyperparameter range for the selection of the best GRU and LSTM models.

Units	Epoch	Batch Size	Dropout	Activation Function	Optimizer
(50, 100, 150)	(50, 100, 150)	(8, 16, 32)	(0.2)	(‘ReLU’)	(‘adam’)

**Table 3 nutrients-15-01199-t003:** Hyperparameter range for the selection of the best Transformer model.

Head Size	Num Heads	Epoch	Batch Size	Dropout	Activation Function	Optimizer
(64, 128, 256)	(2, 4, 8)	(50, 100, 150)	(8, 16, 32)	(0.2, 0.25)	(‘ReLU’)	(‘adam’)

**Table 4 nutrients-15-01199-t004:** Tukey test results: Diff represents the mean difference between the pair of groups; Lower and Upper the lower and upper difference between the pair of groups, respectively; q-value is a value that provides a means to control the positive false discovery rate (pFDR); and *p*-value is the probability of obtaining test results at least as extreme as the result observed, under the assumption that the null hypothesis is correct.

Group 1	Group 2	Diff	Lower	Upper	q-Value	*p*-Value
SARIMAX	LSTM	0.458859	0.152600	0.765117	5.706676	0.001490
SARIMAX	GRU	0.467519	0.161261	0.773778	5.814384	0.001196
SARIMAX	Transformer	0.397567	0.091309	0.703826	4.944415	0.006675
LSTM	GRU	0.008660	−0.297598	0.314919	0.107708	0.900000
LSTM	Transformer	0.061291	−0.244967	0.367550	0.762261	0.900000
GRU	Transformer	0.069952	−0.236307	0.376210	0.869969	0.900000

**Table 5 nutrients-15-01199-t005:** Time costs for the training and forecasting (of 7 days) mean for each model, averaged based on 10 participants.

Model	Computational Time	Standard Deviation	Retraining Time	Forecasting Time
SARIMAX	1.83 s	1.06 s	1.56 ± 1.05 s	0.29 ± 0.03 s
LSTM	13.5 s	3.60 s	12.6 ± 3.30 s	0.92 ± 0.33 s
GRU	12.7 s	1.42 s	12.0 ± 1.22 s	0.86 ± 0.26 s
Transformer	48.6 s	10.7 s	47.9 ± 10.7 s	0.85 ± 0.13 s

**Table 6 nutrients-15-01199-t006:** In this table, the overall results are summed up for each model, reporting each mean value and its associated SD.

Model	RMSE	$RMSE_{reduced}$	Computational Time	Retraining Time
SARIMAX	0.85 ± 0.37	1.95 ± 2.30	1.83 ± 1.06 s	1.56 ± 1.05 s
LSTM	0.39 ± 0.18	0.48 ± 0.24	13.5 ± 3.60 s	12.6 ± 3.30 s
GRU	0.38 ± 0.16	0.45 ± 0.30	12.7 ± 1.42 s	12.0 ± 1.22 s
Transformer	0.45 ± 0.25	0.66 ± 0.28	48.6 ± 10.7 s	47.9 ± 10.7 s

## Data Availability

Data and codes are available upon reasonable request.

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
