# Peer review of "Putting the Personalized Metabolic Avatar into Production: A Comparison between Deep-Learning and Statistical Models for Weight Prediction"

_nutrients, 2023, doi:10.3390/nu15051199_

Round 1
Reviewer 1 Report
This is an interesting followup to the authors' recent publications (references 11 and 12). It would be valuable to use this technology in association with other methodologies to develop models predicting more sensitive and clinically important metrics including blood sugar, serum lipids, and blood pressure, etc
Author Response
We thank the reviewer for their kind comments. We agree on the value of connecting the personalized metabolic avatar with clinical metrics. Indeed, they will be included in the PMA in future works.
Reviewer 2 Report
Please read the attachment. Thank you.

Author Response
Dear Reviewer,
Thank you for taking the time to review our manuscript.
